# Research on the mechanism of pore structure on water transportation in cement-based materials

**Enze Hao**[1,2¤], **Yuhang Li**[1,2¤], **Dali Zhang**[1,2¤*], **Wenbang Zhu**[1,2¤], **Ruiming Liu**[1,2¤], **Xinjie Wang**[1,2¤], **Yali Cao**[1,2], **Yali Gu**[1,2¤], **Xiumei Zheng**[1,2¤]

**1** College of Civil Engineering, Kashi University, Kashi, Xinjiang Uygur Autonomous Region, China,
**2** Xinjiang Key Laboratory of Engineering Materials and Structural Safety, Kashi, Xinjiang Uygur Autonomous Region, China

¤ Current Adress:College of Civil Engineering, Kashi University, Kashi, Xinjiang Uygur Autonomous Region, China
* zhangdali@ksu.edu.cn

## Abstract

To explore the mechanism of water transport in cement-based materials with different water-cement ratios under the influence of pore structure, three types of specimens with different water-cement ratios, namely paste, mortar and concrete, were fabricated. The capillary water absorption and isothermal adsorption-desorption water transport characteristic values of the three types of cement-based materials were studied, and the pore changes and porosity were analyzed. The test results show that for the cement-based materials with the same water-cement ratio, the water absorption and water absorption rate of the paste are greater than those of the mortar and concrete, and the porosity of the paste is greater than that of the concrete and mortar. Based on Darcy's law, a water transport model based on different porosity parameters was established. The results show that the water transport speed of the paste is greater than that of the mortar and concrete, which is in good agreement with the experimental results.

## 1. Introduction

Cement-based materials are the most widely used materials in construction in the world, which have the advantages of convenient materials, low cost and good fire resistance [1,2], and are widely used in houses, Bridges, roads and other places [3–5]. The deterioration of the durability of cement-based materials in harsh environments is usually caused by the presence of water [6,7]. The moisture content and water transport in cement-based materials will affect the physical properties and durability of cement-based materials such as shrinkage, cracking, ion migration, carbonization, corrosion, water absorption and alkali aggregate reaction [8–14]. A large number of studies have been conducted on the influence of water or humidity on the durability of cement-based materials [15–17]. For example, Bohris et al. [18] used

---

---

**Data availability statement:** All relevant data are within the manuscript and its Supporting information files.

**Funding:** This work was financially supported by the Kashi University-level project (202422930, 202422926).

**Competing interests:** The authors have declared that no competing interests exist.

nuclear magnetic resonance technology to study the hydration and pore evolution of silicate cement slurry, and analyzed the proportion of water forms such as capillary water and gel water. Isgor et al. [19] proposed the coupling effect of water and carbon dioxide transport in concrete, and the results showed that water can accelerate concrete carbonization. Homan et al. [20] discussed the influence of water transport on chloride ion permeability in unsaturated concrete, and proposed a revised governing equation. Wang et al. [21] reviewed the nondestructive testing technology capacitance testing to study water distribution in cement-based materials. In these cases, water is either a major factor in the deterioration of the concrete or a transport medium for the aggressive material. Therefore, it is of great significance to study the water transport and distribution in concrete, which can provide a basis for analyzing the durability evolution of concrete structures under erosion environment.

The transport mechanism of water in gelled materials is mainly based on three processes: penetration, diffusion and absorption. The transfer process of water in concrete materials is very complex and depends to a large extent on micro-pore structure, concrete surface, pore size distribution, water content, temperature, etc. [22–25]. Liu et al. [26] used molecular dynamics simulation to reveal the migration mechanism of ions in unsaturated CSH gel pores, and came to the conclusion that migration ability was positively correlated with saturation. Zhang et al. [27] studied the effect of freeze-thaw cycle on capillary water absorption and chloride penetration in ordinary concrete and aerated concrete with different water-cement ratios, and concluded that freeze-thaw cycle accelerated capillary water absorption and chloride penetration. Liu et al. [28] used low-field nuclear magnetic resonance technology to study the water transport rule in ultra-low water-cement ratio mortar under the coupling condition of osmotic pressure and temperature. The results show that the influences of different factors on water transport are osmotic pressure, confining pressure and temperature. Velardo et al. [29] carried out a study on the influence of recycled aggregate on water transfer of concrete, and the results showed that recycled aggregate concrete reduced water permeability by 64%, absorbance by 17% and oxygen permeability by 58%. Yu et al. [30] studied the effects of polyacrylic acid, polyvinyl alcohol and polyethylene glycol on the durability of modified cement-based materials. The results show that polyethylene glycol has the best water blocking effect, followed by polyacrylic acid, and polyvinyl alcohol is the most unfavorable.

The construction of water transport model is the main means to explore the internal mechanism of cement-based materials and predict the change of properties. In addition to studying the influencing factors of water transport in concrete, researchers also proposed some water transport models, such as finite element model, uniform model, multi-scale model, three-dimensional unsaturated water transport model, etc. [31–35]. For example, Zhang et al. [36] used the convection-diffusion water transport model coupling to study the influence of boundary conditions on simulated water transport. Bao et al. [37] established a prediction model of water transport in load-induced damaged concrete based on unsaturated flow theory and Mazar damage variables. Srimook et al. [38] used the mesoscale rigid body spring model to study abnormal water transport in cement-based materials, and found that the anomalies

of penetration depth and cumulative water absorption showed an S-shaped curve. Min et al. [39] established a coupling model considering temperature and humidity based on heat gradient and humidity gradient coefficient. Combined with the experiment, it was found that temperature and humidity accelerated the transport of water and accelerated the destruction of cement-based materials.

In conclusion, there are more studies on water transport in concrete, but few studies on water transport in cement-based materials with multi-scale pore structure. Therefore, it is very important to study the quantitative effect of pore structure on water transport in cement-based materials. In view of this, this study considered the capillary effect, porosity, adsorption desorption and other effects, based on scanning electron microscopy, compression and nuclear magnetic resonance technology, quantitative analysis of the effect of porosity on the water transport of cement-based materials, and explored the establishment of a numerical model considering the porosity, to provide a reference for the durability design of cement-based materials under long-term corrosion environment.

## 2. Materials and methods

### 2.1. Raw materials

The P·O 42.5 cement used in this study was produced by Xinjiang Tianshan Cement Co.The chemical composition was analyzed by X-ray fluorescence spectrometer (XRF), and the chemical composition is detailed in Table 1. The mixing water is tap water; The superplasticizer is a polycarboxylic acid superplasticizer, liquid, and the water reduction rate is 25% to 40%. The fine aggregate is washed sand with a fineness modulus of 2.02, which is fine sand with an apparent density of 2460 kg/m$^3$. Coarse aggregate is broken gravel with a particle size of 5–10 mm continuously graded with a mud content of 0.7% and an apparent density of 2680 kg/m$^3$.

### 2.2. Mix proportion

In this experiment, specimens of paste(P), mortar(M) and concrete(C) with three water-cement ratios were designed. The mix proportion design was carried out in accordance with the specifications "Code for Design of Mix Proportions of Masonry Mortar" [40] (China JGJ/T 98–2010) and "Code for Design of Mix Proportions of Ordinary Concrete" [41](China JGJ55–2011). The mix proportions are shown in Table 2.

**Table 1. Chemical composition of cement (Mass percentage/%).**

| Ingredient | SiO$_2$ | Al$_2$O$_3$ | Fe$_2$O$_3$ | CaO | MgO | SO$_3$ | K$_2$O | Na$_2$O | Li$_2$O | Others |
|---|---|---|---|---|---|---|---|---|---|---|
| Cement | 20.12 | 5.75 | 3.26 | 63.44 | 0.98 | 2.71 | 0.49 | 0.73 | 2.13 | 0.39 |

**Table 2. Mix ratio of cement-based materials.**

| Number | Water-binder ratio | Amount of material (kg·m$^{-3}$) | | | | |
|---|---|---|---|---|---|---|
| | | Water | Cement | Sand | Stone | Superplasticizer |
| P1 | 0.35 | 508 | 1451 | 0 | 0 | 2.9 |
| P2 | 0.42 | 579 | 1380 | 0 | 0 | 2.1 |
| P3 | 0.50 | 653 | 1306 | 0 | 0 | 1.3 |
| M1 | 0.35 | 219 | 627 | 1254 | 0 | 3.1 |
| M2 | 0.42 | 237 | 564 | 1297 | 0 | 2.0 |
| M3 | 0.50 | 250 | 499 | 1348 | 0 | 1.2 |
| C1 | 0.35 | 156 | 448 | 896 | 896 | 2.2 |
| C2 | 0.42 | 176 | 419 | 964 | 838 | 1.5 |
| C3 | 0.50 | 210 | 386 | 1044 | 774 | 0.9 |

## 2.3 Test methodology

The specimen is a cubic body with dimensions of 100 mm × 100 mm × 100 mm, as shown in Fig 1(a). To prevent the influence of incomplete cement hydration on the test results, after standard curing for 90 days, the specimen was divided into two parts using a cutting machine, and then placed in an 80°C oven to dry until the mass no longer decreased. Subsequently, epoxy resin was used to seal the five sides except for the cutting surface. The cutting machine is shown in Fig 1(c). The exposed surface of the prepared sample was placed on a pre-set support in a plastic basin, and water was added until the liquid level was about 5 mm above the soaking surface of the specimen, and water absorption test was carried out according to ASTM C1585, as shown in Fig 1(b). After drying the surface moisture of the specimens at time points of 5 minutes, 10 minutes, 20 minutes, 30 minutes, 1 hour, 2 hours, 3 hours, 4 hours, 5 hours, 6 hours, 10 hours, 14 hours, and 24 hours, weigh the specimens again. Then, continue weighing every day until the weight no longer changes. The mass changes of cement-based material samples in each group were recorded at different times. The average value of mass changes of 3 test blocks in each group was taken as the final result, and the relationship between time and water transport quantity was drawn.

Different supersaturated salt solutions will reach different humidity levels in a closed space. In this experiment, based on the hygrothermal performance of building materials and products [42], $MgCl_2$, NaBr, NaCl, KCl, and $K_2SO_4$ were selected to prepare supersaturated salt solutions. The humidity levels that the five solutions can achieve are shown in Table 3. Humidity sensors were placed inside the humidity box, and the sensors were connected to an external USB-to-485 module. The box opening was covered with plastic film to isolate the air, and the humidity changes were observed through a computer port. When the humidity inside the humidity box reached a stable state, adsorption and desorption tests were conducted. The adsorption and desorption specimens were 10 mm × 100 mm × 100 mm thin slices, as shown in Fig 1(d). After the environmental humidity in the humidity box stabilized, the prepared dry specimens and

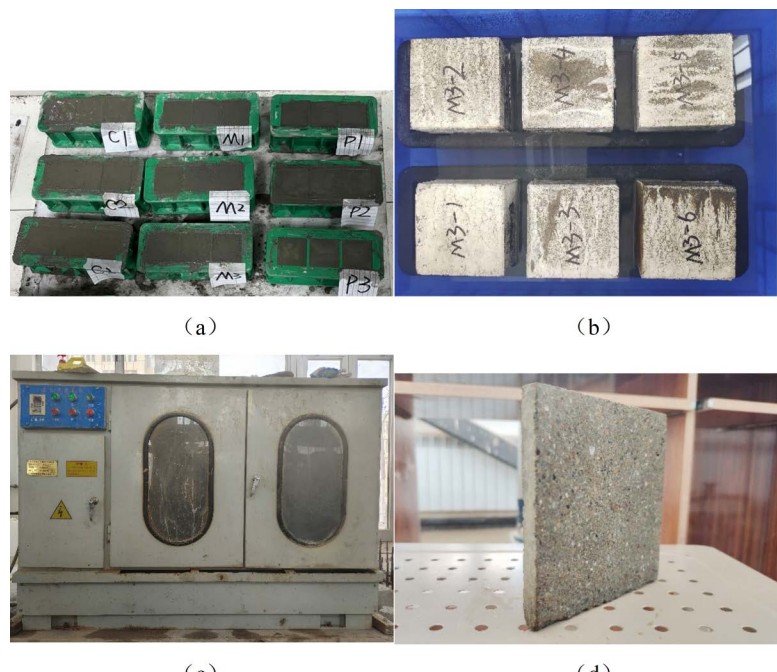

（a）　　　　　　　　　　　（b）

（c）　　　　　　　　　　　（d）

**Fig 1. Sample preparation and testing process: (a) Preparation of specimens; (b) Water absorption test; (c) Cutting machine; (d) 10 mm mortar specimens.**

**Table 3. Humidity corresponding to saturated salt solution [42].**

| Saturated salt solution | MgCl$_2$ | NaBr | NaCl | KCl | K$_2$SO$_4$ |
|---|---|---|---|---|---|
| Humidness | 33 | 58 | 76 | 86 | 98 |

saturated specimens were placed inside. After three months of adsorption and desorption tests, the slices were taken out and weighed every week. Three specimens were weighed each time until the mass of the specimens no longer changed, and the final saturation was calculated.The saturation calculation formula is as follows [43]:

$$\Theta = \frac{m - m_d}{m_s - m_d}$$

(1)

where:$\Theta$represents the saturation of the specimen, m denotes the mass of the specimen, $m_d$ indicates the mass of the specimen in the dry state, and $m_s$ represents the mass of the specimen when saturated.

## 2.4. Test method

SEM (scanning electron microscopy) test: Using the PhenomProx scanning electron microscope produced by Fuxian Science Instrument Co., Ltd., as shown in Fig 2(a).The specimens cured to the specified age were crushed, and flat thin slices with a diameter of less than 10 mm and a thickness of approximately 3–4 mm were taken, put into a container and anhydrous ethanol was added to stop hydration. Then they were placed in an oven at 40°C for 24 hours. The dried thin slices were placed on the sample stage, sent to the vacuum gold spraying platform for gold spraying treatment, and then placed in the electron microscope equipment for observation and photography.

 MIP (Mercury Intrusion Porosimetry) test: The AutoPore IV 9500 mercury porosimeter produced by Micromeritics Instrument Corporation of the United States was used. The measurement of pores within a specific pore size range was achieved through staged pressure application. The instrument is shown in Fig 2(b). Samples with a diameter of less than

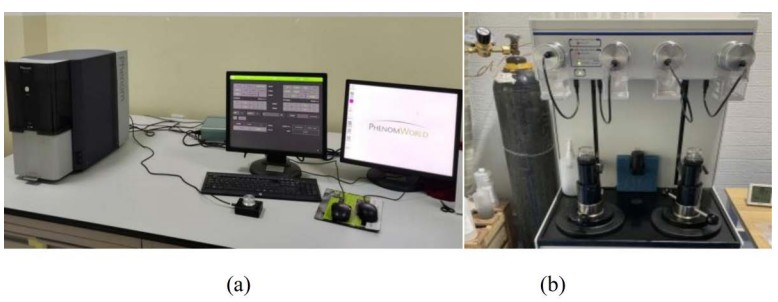

(a) (b)

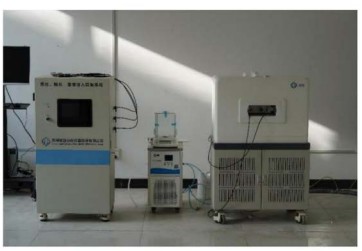

（c）

**Fig 2. Microscopic instruments and test samples:(a)scanning electron microscopy;(b)Mercury Intrusion Porosimetry;(c)High precision magnetic resonance concrete micro analysis.**

10 mm and a thickness of approximately 5 mm were taken from the reserved specimens. Firstly, the specimens were placed in anhydrous ethanol for 48 ± 0.5 hours to stop hydration. The pore changes of specimens after different erosion ages and different concentrations of erosion were tested. The contact angle was 130°, and the pore size measurement range was 0.003–360 µm.

NMR (Nuclear Magnetic Resonance) test: The MesoMR12-060H-I high-precision magnetic resonance concrete microstructure analyzer produced by Suzhou Niumai Analytical Instrument Co., Ltd., China, was used. Specimens of 40 mm × 40 mm × 40 mm were prepared and standard cured for 90 days. Then, the specimens were placed in a vacuum water retention machine. After water retention was completed, the pore changes of specimens with different water-cement ratios were tested. The pore size measurement range was 0.0002 to 200 µm.

## 3. Results and analysis

### 3. 1 Microscopic analysis

**3.1.1. MIP analysis.** In the mercury intrusion porosimetry test, liquid mercury penetrates into the pores of the material under pressure. Once the pressure application concludes, a portion of the liquid mercury remains within the pores [44]. Consequently, the volume of mercury intrusion exceeds that of mercury extrusion, with the residual portion being ink-bottle pores. The pores that can freely release mercury after unloading are regarded as effective pores. Effective pores serve as the primary channels for the transmission of water and erosive media and are pivotal in determining the durability of the material. Figs 3(a), 3(b), and 3(c) present the mercury intrusion and extrusion curves for P, M, and C. It is observable from the figures that for cement-based materials with varying water-cement ratios, the volume of mercury intrusion is greater than that of mercury extrusion. The trend of cumulative mercury intrusion volume reduction is P1 > P2 > P3, M1 > M2 > M3, and C1 > C2 > C3, while the cumulative mercury extrusion volume undergoes negligible change. This phenomenon occurs because as the water-cement ratio increases, the pore structure of the cement-based material becomes more intricate, and the pore size distribution range widens, resulting in a greater number of ink-bottle pores.

Figs 3(d), 3(e), and 3(f) depict the pore size distribution curves for P, M, and C. It can be discerned from the figures that for cement-based materials with distinct water-cement ratios, there are substantial disparities in pore distribution. The porosities of P1, P2, P3, M1, M2, M3, C1, C2, and C3 obtained through the mercury intrusion test are 21.73%, 26.92%, 31.45%, 17.64%, 17.94%, 19.32%, 9.61%, 9.67%, and 12.02%, respectively. With the incorporation of sand and gravel, the porosity undergoes a significant reduction, and the pore alterations are pronounced in the range of 0 nm to 1000 nm and beyond 100,000 nm. Figs 3(g), 3(h), and 3(i) illustrate the relationship curves between the cumulative mercury intrusion volume and pore size for P, M, and C. It is evident from the figures that the cumulative mercury intrusion volume distribution is P3 > P2 > P1, M3 > M2 > M1, and C3 > C2 > C1.

As shown in Figs 3(j) and 3(k), they are the pore size distribution diagrams of P, M, and C. According to the classification of internal pore diameters of cement-based materials by Academician Wu Zhongwei [45]: the first level is harmless pores, with diameters ranging from 0 nm to 20 nm, usually in a closed or isolated state, not forming a connected network, hardly reducing the mechanical strength of the material, and having no significant contribution to permeability; the second level is slightly harmful pores, with diameters ranging from 20 nm to 50 nm, which may form local connected paths, but the overall permeability is low, slightly weakening the strength, allowing a small amount of water or ions to slowly penetrate, but the threat to durability is limited; the third level is harmful pores, with diameters ranging from 50 nm to 200 nm, which are prone to form continuous channels, significantly increasing the permeability of the material, obviously reducing the density and mechanical strength, and accelerating the invasion of harmful substances (such as chloride ions and sulfates); the fourth level is highly harmful pores, with diameters greater than 200 nm, large pores or crack-level pores, usually caused by construction defects or shrinkage stress, seriously damaging the strength and impermeability of the material, becoming stress concentration points and crack propagation paths, and significantly shortening the service life

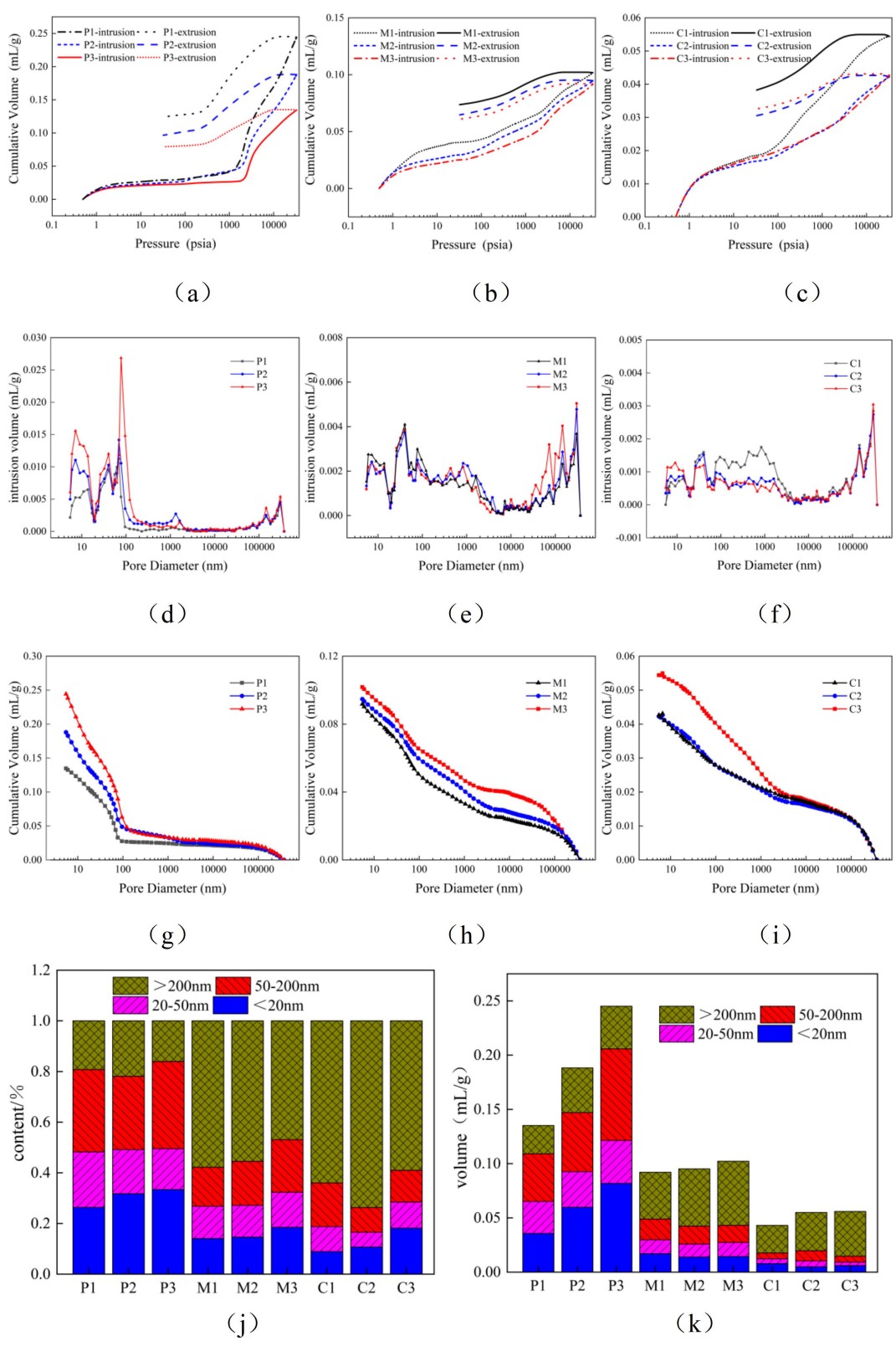

（a）　　　　　　　　　　（b）　　　　　　　　　　（c）

（d）　　　　　　　　　　（e）　　　　　　　　　　（f）

（g）　　　　　　　　　　（h）　　　　　　　　　　（i）

（j）　　　　　　　　　　　　　　　　（k）

**Fig 3. MIP:(a) Mercury curve of P; (b) Mercury curve of M; (c) Mercury curve of C; (d)Pore size distribution curves of P; (e) Pore size distribution curves of M; (f) Pore size distribution curves of C; (g) P cumulative tribute and aperture size relationship curve; (h) M cumulative tribute and aperture size relationship curve; (i) C cumulative tribute and aperture size relationship curve; (j, k) P, M, C pore change accumulation map.**

of the structure. It can be seen from the figures that the porosity of cement-based materials with different water-cement ratios varies greatly, showing that the porosity increases continuously with the increase of the water-cement ratio. This is because compared with mortar and concrete, the paste is composed of cement and water, and its internal pore size distribution is more uniform, with a higher proportion of harmless and slightly harmful pores. However, the addition of fine aggregates in mortar reduces the porosity but increases the proportion of harmful and highly harmful pores. With the addition of coarse aggregates, the proportion of harmful and highly harmful pores increases, but the reduction in porosity helps to increase the durability of cement-based materials.

It can be seen that the complexity of the pore structure in cement-based materials increases mainly due to the higher water-cement ratio, which leads to the formation of more fine pores and capillary pores. These pores play a role in storing water and aggressive media in cement-based materials, but they also reduce the durability of the materials. In cement-based materials, the size and distribution of pores directly affect the rate and range of water transmission. Small pores can restrict the rapid flow of water, while large pores may become fast channels for water transmission. Therefore, an increase in porosity, especially the increase in harmful pores and multi-harmful pores, makes the material more suscepti-ble to erosion by aggressive media, thereby affecting its long-term structural stability and durability.

**3.1.2. Nuclear magnetic resonance pore testing and analysis.** The T2 spectrum of nuclear magnetic resonance (NMR) reflects the distribution of pore sizes within cement-based materials. The shorter the transverse relaxation time T2, the smaller the pore radius; the longer the T2, the larger the pore radius. The position of each peak in the T2 spectrum distribution curve is related to the pore size, and the size of the integral area of each peak reflects the change in the number of pores within the cement-based material [46].

Fig 4 shows the T2 spectrum distribution curves of 90dP, M and C NMR for curing. As can be seen from the figure, the NMR T2 spectra of P1, P2 and P3 mainly present a "main peak" structure. The main peaks are distributed in the regions with short transverse relaxation time, and the signal intensity is P3>P2>P1, indicating that the number of small pores in

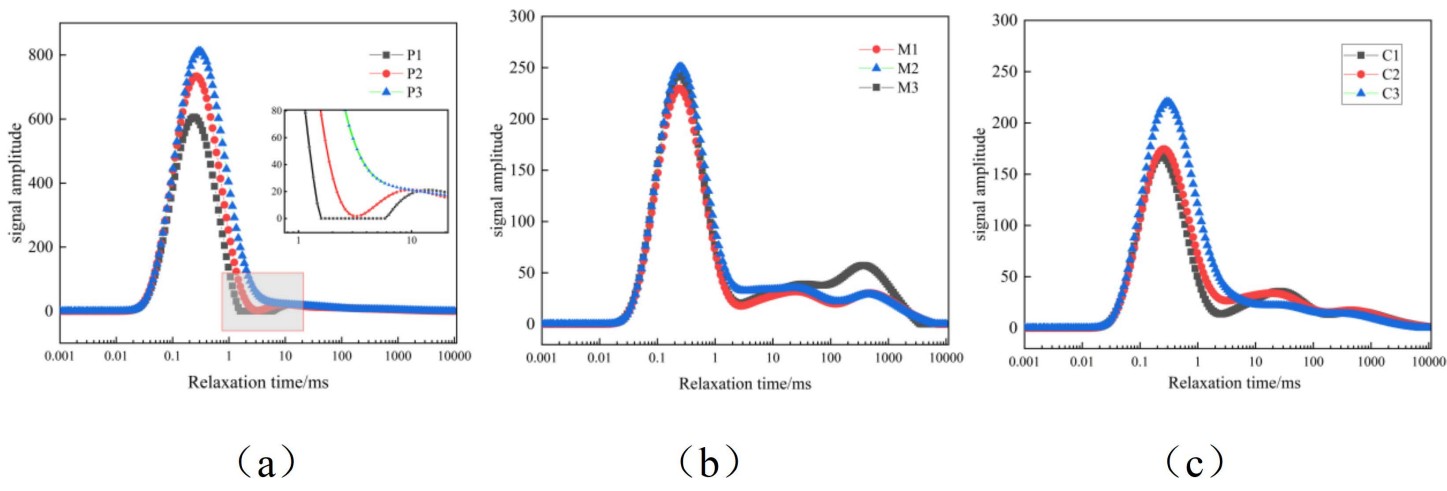

**Fig 4. Curve of NMR T2 Spectrum Analysis: (a) 90d, P (b) 90d, M (c) 90d, C.**

P1, P2 and P3 is mostly, and only a small part is large pores.The signal peak of the paste specimens shows an increasing trend as the water-cement ratio increases, indicating that the larger the water-cement ratio, the more pores there are.

The T2 spectra of M1, M2, M3, C1, C2, and C3 nuclear magnetic resonance mainly present a "primary and secondary peak" structure. The primary peaks are all distributed at the shorter transverse relaxation time, and their signal intensity and the area under the distribution curve are significantly greater than those of the secondary peaks, indicating that the number of small pores inside the mortar and concrete accounts for a larger proportion, while only a small part are large pores. This is because fine aggregates and coarse aggregates that can increase the internal cohesion of the matrix are added during the mixing process of mortar and concrete, thereby improving their pore structure. The improvement of the pore structure of cement-based materials is closely related to the composition and preparation process of the materials. During the preparation process, the proportion of raw materials such as cement, water, sand, and stone, as well as the process conditions such as mixing and curing, will all affect the pore structure of concrete. The porosity of the paste specimens is relatively large, and as sand and stone are added, the number of pores in the cement-based materials gradually decreases, and the porosity gradually reduces.

As shown in Fig 5, it is a cumulative distribution graph of the pore changes in the paste, mortar and concrete. The proportion of various types of pores such as P1, P2, P3, M1, M2, M3, C1, C2 and C3 is statistically described based on the T2 spectrum distribution curve of nuclear magnetic resonance. It can be seen from the figure that for the paste, mortar and concrete specimens, as the water-cement ratio increases, the porosity continuously increases, the proportion of harmless and slightly harmful pores decreases, and the proportion of harmful and highly harmful pores increases. With the addition of sand and gravel, the influence of the water-cement ratio on porosity decreases and the porosity is reduced. The increase in the water-cement ratio leads to changes in the internal pore structure of cement-based materials, mainly because more water fails to participate in the hydration process of cement, thus forming more pores inside the material. These pores are mainly manifested as an increase in harmful and highly harmful pores in size, which directly affects the mechanical properties and durability of the material. The reduction of harmless and slightly harmful pores means a decrease in micro-pores inside the material, which to some extent is beneficial to improving the density of the material. However, overall, the increase in harmful and highly harmful pores has a more significant negative impact on the material's performance. In addition, the addition of sand and gravel to some extent improves the pore structure of the material. As aggregates, sand and gravel can fill some pores and reduce the formation of harmful and highly harmful pores, thereby

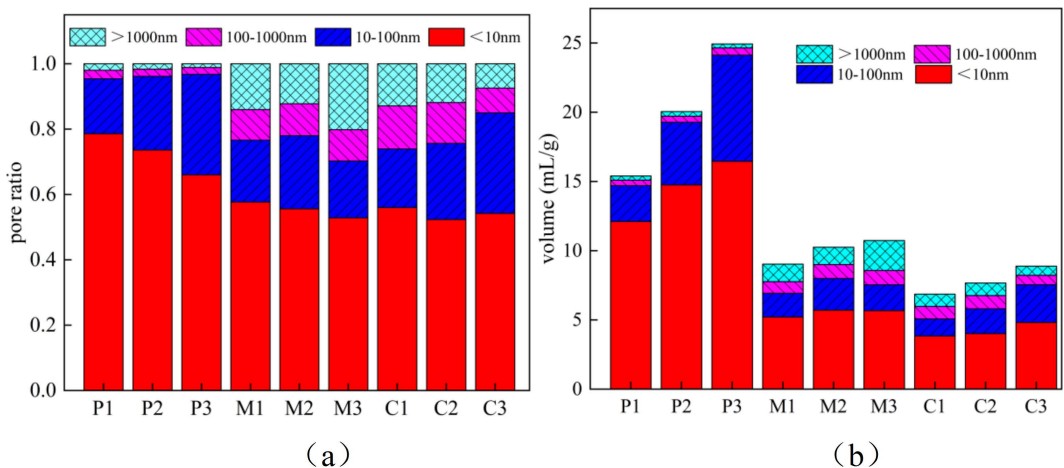

**Fig 5. P. M and C pore distribution: (a) Pore distribution (b) pore change.**

improving the density and overall performance of the material. However, this improvement effect is limited when the water-cement ratio is too high, as excessive water still leads to an increase in porosity.

The pore changes in cement-based materials were analyzed by combining the mercury intrusion method and nuclear magnetic resonance (NMR) technology. It was found that there were certain differences in the porosity measured by the two methods, but the changing trends were consistent. This was mainly due to the different sample sizes tested by the two methods, which led to minor differences in the measurement results. The mercury intrusion method is usually suitable for smaller volume samples and can more accurately reflect the local pore structure, while NMR technology is suitable for larger volume samples and can provide more comprehensive pore distribution information. Despite the differences, both methods indicated that with the increase of water-cement ratio, the harmful and multi-harmful pores in cement-based materials significantly increased, which was consistent with the changing trend of water absorption of the materials. By comprehensively analyzing the test results of the two methods, a more comprehensive understanding of the influence mechanism of pore structure on the performance of cement-based materials can be achieved, providing important references for optimizing material ratios and improving material performance.

**3.1.3. SEM analysis.** The experiment utilized a scanning electron microscope to observe the specimens P1, P2, P3, M1, M2, M3 at a magnification of 9900 times after 90 days of curing, and the internal microstructure of C1, C2, C3 at a magnification of 9500 times, as shown in Fig 6. It can be seen from the figure that the microstructure varies significantly with different water-cement ratios. For specimens P1, M1 and C1, after 90 days of curing, the internal structure is relatively smooth with a certain number of pores, but the pores are few. For specimens P2, M2 and C2, after 90 days of curing, the internal structure is relatively rough with an increased number of pores. For specimens P3, M3 and C3, after 90 days of curing, there are a large number of pores inside. This is because an increase in the water-cement ratio has a certain impact on the hydration degree of cement-based materials. Under a higher water-cement ratio, the excess water in the cement paste evaporates during the hardening process, leaving more pore space. These pores not only affect the mechanical properties of the material but also have a negative impact on its durability and impermeability. From the perspective of microstructure, the size and distribution of the pores are uneven, which may be related to the morphology and distribution of the hydration products during the cement hydration process. An increase in the water-cement ratio causes the distance between the hydration products to increase, leading to the formation and expansion of pores. In addition, the shape and connectivity of the pores also have an important influence on the water transmission characteristics of the material.

## 3.2. Effect of porosity on capillary water absorption

The variation of water transport characteristics during the experiment is expressed by the change in the water absorption per unit area and the initial water absorption coefficient S. The variation of water transport quality with time is expressed by the following equation [47]:

$$I = \frac{m}{\rho_l A} = \varphi \left( \frac{r\gamma}{2\eta_l} \cos\theta \right)^{1/2} t^{1/2} = S \cdot t^{1/2}$$

(2)

Where:$I$ water absorption per unit area; A cross-sectional area of the specimen's water absorption surface;$\rho_l$ density of water;$\varphi$ porosity of the material;$r$ radius of the capillary;$\gamma$ surface tension of water;$\eta$ viscosity coefficient of water;$\theta$ contact angle between water and the capillary;$t$ water absorption time;$S$ initial water absorption coefficient.

The unit area water absorption capacity is expressed by the following formula [48]:

$$W = \left( \varphi\rho \sqrt{\frac{r\gamma_{lv}\cos\theta}{2\mu}} \right) \cdot \sqrt{t}$$

(3)

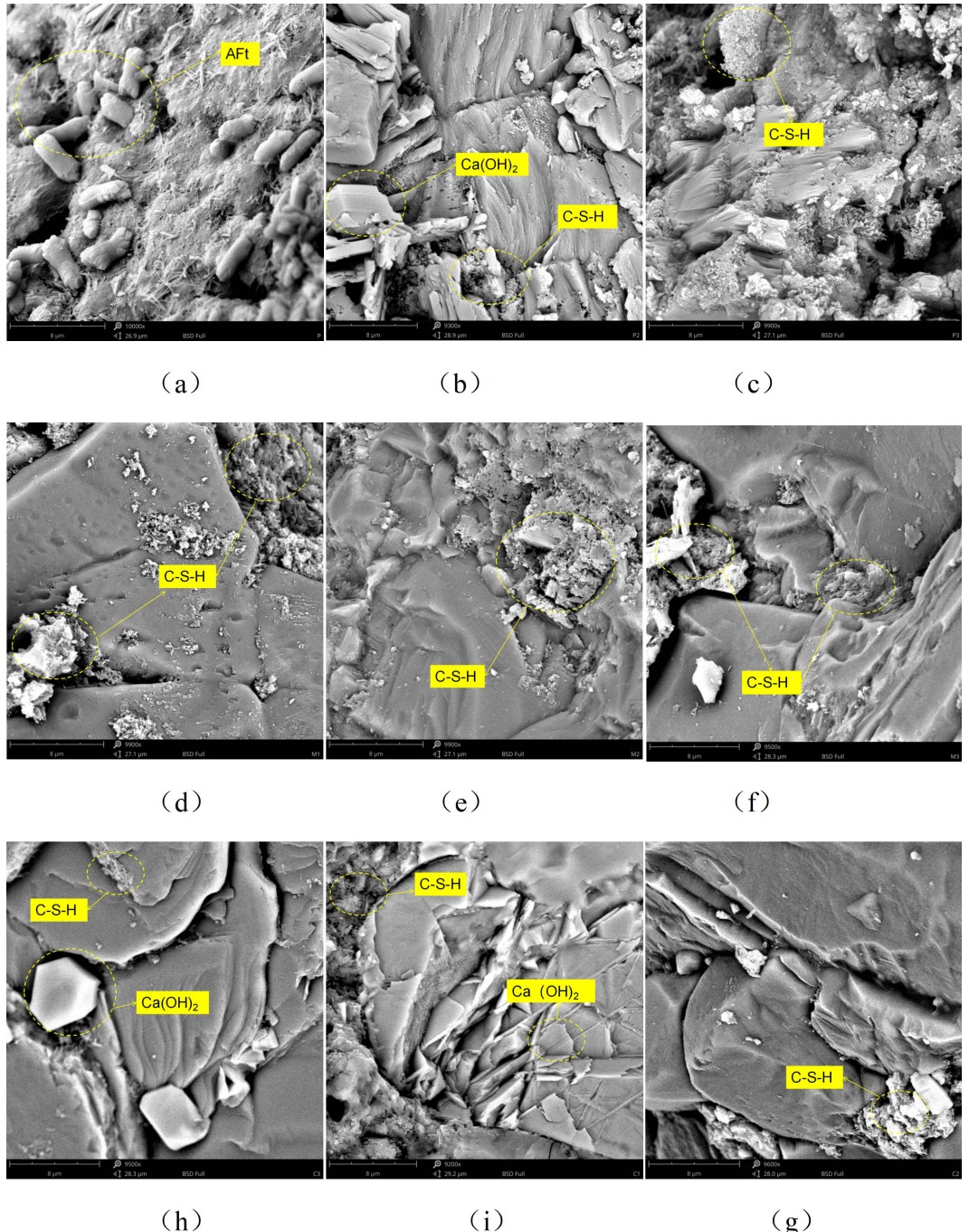

**Fig 6. SEM of cement-based materials with different water-cement ratios: (a) 90d, P1 (b) 90d, P2 (c) 90d, P3 (d) 90d, M1 (e) 90d, M2 (f) 90d, M3 (d) 90d, M1 (e) 90d, M2 (f) 90d, M3.**

Where:$w$ represents the water absorption per unit area;$\varphi$ is the porosity;$\rho$ is the density of water;$r$ is the radius of capillary pores;$\gamma_{lv}$ is the surface tension at the gas-liquid interface;$\theta$ is the contact angle, which is generally 0;$\mu$ is the liquid viscosity coefficient;$t$ is the time.

Fig 7 shows the water absorption test results of pastes, mortars, and concretes with different water-cement ratios. It can be seen from the figure that the water absorption of P1 reaches saturation at 12,000g/m², that of P2 at 15700g/m², and that of P3 at 16300g/m². The water absorption of M1 reaches saturation at 4,000g/m², that of M2 at 6,000g/m², and that of M3 at 8,000g/m². The water absorption of C1 reaches saturation at 3,000g/m², that of C2 at 4,000g/m², and that of C3 at 6,000g/m². The water absorption of specimens shows that the water absorption of pure cement paste is greater than that of mortar, which in turn is greater than that of concrete. This is because as the amount of fine aggregate and coarse aggregate increases, the porosity decreases, and the ability of cement-based materials to resist water infiltration is enhanced. The addition of aggregates optimizes the pore structure through the filling effect, reduces water absorption, and thereby improves the impermeability and durability of cement-based materials. This rule provides a theoretical basis for the scientific proportioning, structural design, and durability assessment of engineering materials, which is of great significance for ensuring the safety and economy of infrastructure. The size of porosity directly affects the transmission path and speed of water within the material. When the porosity is high, water is more likely to enter the material through capillary action, resulting in increased water absorption. The reason for this is that the pore structure of cement-based materials is complex and variable. When fine aggregate and coarse aggregate are added, the pore structure of the material becomes more compact, the porosity decreases, and the channels for water transmission are reduced, thus reducing water absorption and enhancing the material's resistance to water infiltration.

The water absorption test results of pastes, mortars, and concretes with different water-cement ratios indicate that as the amount of aggregate increases, the water absorption saturation of the material gradually decreases. This suggests that the addition of aggregate not only reduces the porosity but may also change the connectivity of the pores, making it more difficult for water to enter the material through capillary action. Additionally, the time required for different mix ratios of cement-based materials to reach saturation is similar, indicating that the initial saturation has little impact on the time it takes for the material to reach saturation again when absorbing water. The main factors influencing this are the water absorption coefficient and the total capillary water absorption. Therefore, by adjusting the material's mix ratio, the water absorption performance of cement-based materials can be effectively controlled, thereby improving their durability and service life.

Fig 8 shows the fitting curves of the water absorption of fresh concrete, mortar and concrete with different water-cement ratios. From the figure, it can be seen that for cement-based materials, as the porosity increases, the water absorption also increases continuously. Figure (a) shows the fitting curve of the porosity and water absorption of cement-based

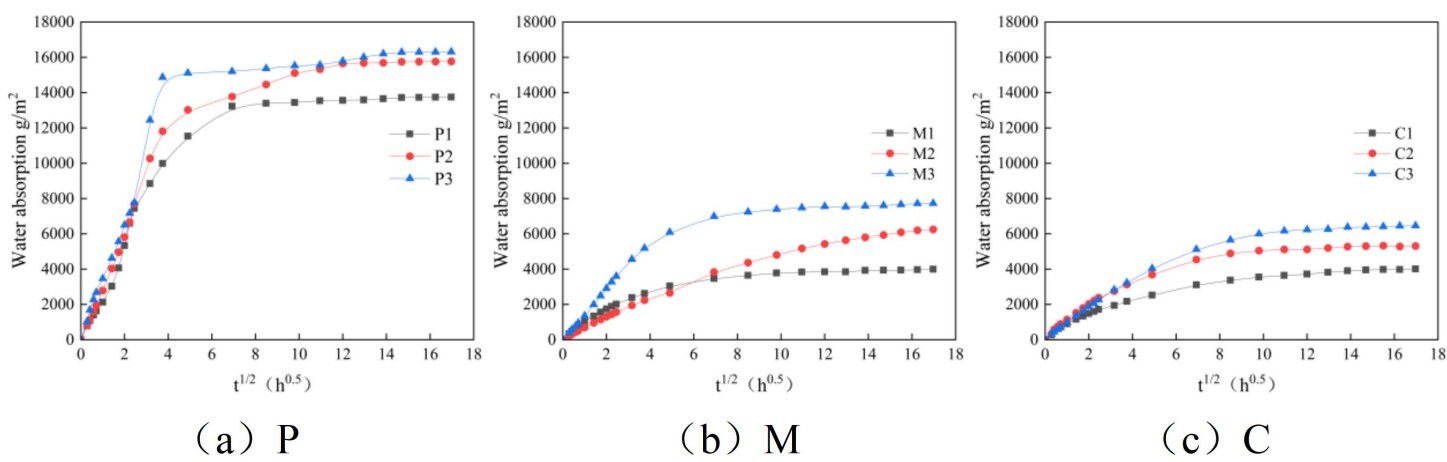

**Fig 7. Water absorption test drawing of clean pulp, mortar and concrete.** (a) P,(b) M, (c) C.

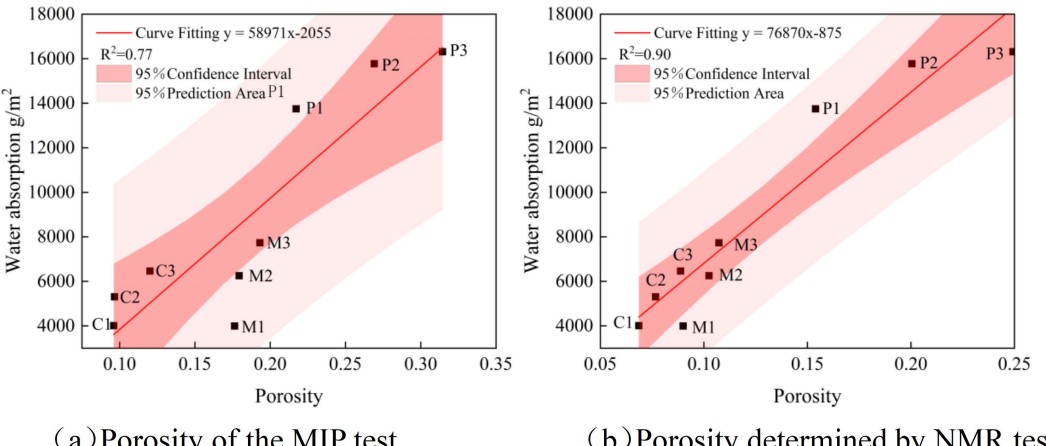

（a）Porosity of the MIP test   （b）Porosity determined by NMR test

**Fig 8. Fitting curves of porosity and water absorption for slurries, mortars, and concrete.** (a) Porosity of the MIP test, (b) Porosity determined by NMR test.

materials tested by MIP, with a relatively large width of the 95% confidence band and prediction band, and a relatively small correlation coefficient of 0.77. Figure (b) shows the fitting curve of the porosity and water absorption of cement-based materials tested by NMR, with a relatively small width of the 95% confidence band and prediction band, and a relatively small correlation coefficient of 0.90. There are certain differences in the porosity measured by the two methods (MIP and NMR), but the trend is consistent.

### 3.5. Isothermal adsorption desorption test

As depicted in Fig 9, it shows the isothermal adsorption-desorption curve of cement-based materials. It can be observed from the figure that under adsorption conditions, when the mass of the specimen reaches equilibrium, the corresponding relationship between saturation and environmental humidity is as follows: at the same environmental humidity, the higher the water-cement ratio, the greater the saturation of the specimen. Under desorption conditions, when the mass of the

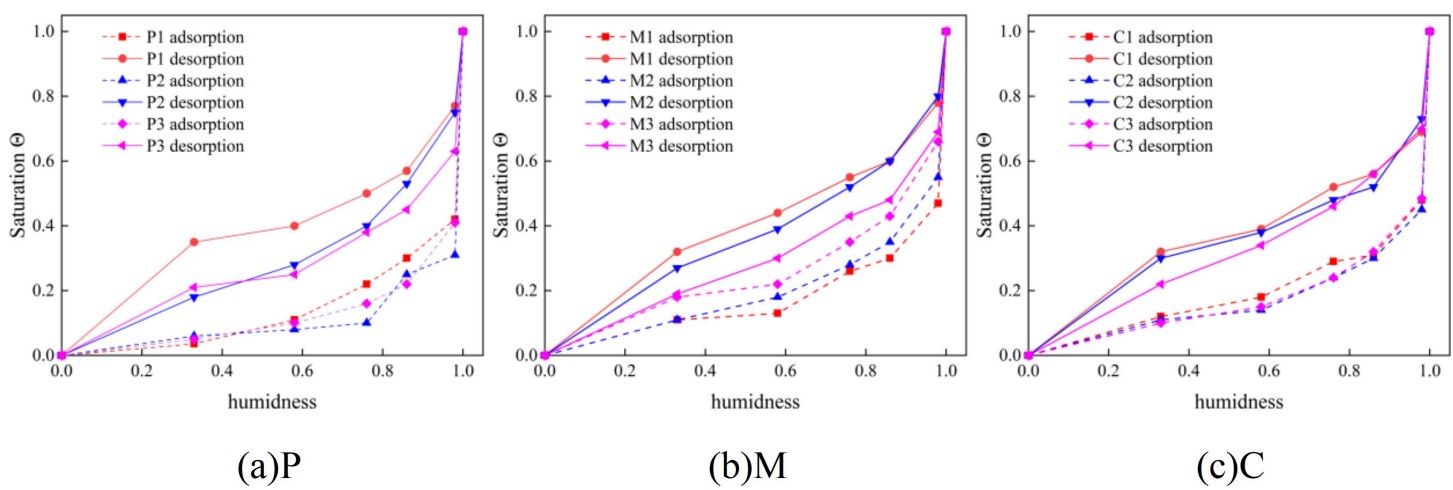

(a)P   (b)M   (c)C

**Fig 9. Isothermal adsorption desorption curve of cement-based materials.** (a) P, (b) M, (c) C.

specimen reaches equilibrium, the corresponding relationship between saturation and environmental humidity is that at the same environmental humidity, the higher the water-cement ratio, the smaller the saturation achieved by the cement-based material in the equilibrium state.

Cement-based materials, being complex porous substances, will exhibit adsorption or desorption phenomena under the effect of capillary adsorption force in different humidity environments [49]. This is because solid materials, in order to reduce the Gibbs free energy of the surface, absorb gas molecules from the external environment. After a certain period, the mass of the porous material reaches stability with a certain degree of saturation. The saturation after adsorption and desorption reflects the fundamental performance of water transmission and retention in cement-based materials and serves as an important basis for studying water transmission within them.

Regarding the adsorption process of cement-based materials, the saturation reached at equilibrium for cement-based materials with different water-cement ratios increases with the increase of the water-cement ratio. This is because the larger the water-cement ratio, the greater the porosity and the larger the pore size. The larger pore size leads to a decrease in the capillary adsorption force of the cement-based material during the adsorption process, thus resulting in a smaller saturation ($\Theta$) at equilibrium.

For the desorption test, a simpler pore path and a larger porosity make it easier for water to evaporate from the cement-based material. In contrast, cement-based materials with low porosity have a complex pore structure, making water evaporation difficult. The complex pores and smaller pore size result in a stronger capillary adsorption force. Therefore, cement-based materials with smaller porosity have a stronger ability to resist water invasion.

Due to the fact that the desorption process of cement-based materials always lags behind the adsorption process, the desorption curve is always above the adsorption curve. Therefore, an isothermal adsorption-desorption curve can form a closed area between the relative humidity of 0 and 1. This closed area is called a hysteresis loop, and the size of the hysteresis loop area reflects the complexity and connectivity of the pore structure of cement-based materials.

The area of the hysteresis loop can be calculated by the following formula:

$$A = \int_0^1 F(x)dx - \int_0^1 f(x)dx \tag{4}$$

In the formula, F(x) represents the desorption fitting curve equation; f(x) represents the adsorption fitting curve equation; 0 and 1 are the environmental humidity conversion ratios.

The adsorption-desorption scatter plot obtained was fitted with a polynomial to obtain the fitting equation: $f(x) = k_1x^3 + k_2x^2 + k_3x + b$. The adsorption-desorption fitting coefficients are shown in Table 4.

After fitting the adsorption-desorption polynomial curve, the hysteresis loop area can be calculated by integration, as shown in Fig 10.

As shown in 10, as the water-cement ratio varies from 0.35 to 0.5, the area of the hysteresis loop continuously decreases. This indicates that with the increase of the water-cement ratio, the pore connectivity and average pore diameter of the cement-based material increase, thereby reducing the capillary retention during the desorption process. This trend is basically consistent with the MIP and NMR analysis. This phenomenon is of great significance for the moisture transport characteristics of cement-based materials under different water-cement ratio conditions, providing a theoretical basis for optimizing material design and improving durability.

## 4. Simulation

The water absorption performance of pure slurry, mortar and concrete was tested by methods such as measuring water absorption and porosity, but there are still some deficiencies. In this paper, the Darcy's Law module of Comsol Multiphysics was used to establish a water transmission model to further reveal the influence of porosity on water absorption.

**Table 4.  Fitting results of adsorption and desorption scatter points.**

| Number | $k_1$ | $k_2$ | $k_3$ | b | $R^2$ |
|---|---|---|---|---|---|
| P1 adsorption | 3.239 | −3.572 | 1.092 | −0.009 | 0.804 |
| P2 adsorption | 4.395 | −5.310 | 1.632 | −0.009 | 0.729 |
| P3 adsorption | 4.284 | −5.447 | 2.205 | −0.011 | 0.796 |
| P1 desorption | 3.486 | −4.925 | 2.356 | −0.003 | 0.972 |
| P2 desorption | 2.773 | −3.255 | 1.393 | −0.004 | 0.972 |
| P3 desorption | 3.412 | −4.323 | 1.766 | −0.004 | 0.920 |
| M1 adsorption | 3.838 | −4.623 | 1.567 | −0.007 | 0.830 |
| M2 adsorption | 3.553 | −4.220 | 1.485 | −0.008 | 0.880 |
| M3 adsorption | 3.560 | −4.375 | 1.685 | −0.005 | 0.939 |
| M1 desorption | 2.600 | −4.623 | 1.964 | −0.004 | 0.971 |
| M2 desorption | 2.418 | −4.220 | 1.653 | −0.003 | 0.980 |
| M3 desorption | 2.758 | −4.375 | 1.497 | −0.006 | 0.941 |
| C1 adsorption | 3.624 | −4.429 | 1.588 | −0.009 | 0.822 |
| C2 adsorption | 3.914 | −4.760 | 1.620 | −0.008 | 0.816 |
| C3 adsorption | 3.776 | −4.515 | 1.530 | −0.008 | 0.844 |
| C1 desorption | 2.870 | −4.032 | 2.038 | −0.003 | 0.937 |
| C2 desorption | 3.317 | −4.567 | 2.146 | −0.005 | 0.953 |
| C3 desorption | 2.354 | −2.919 | 1.448 | −0.004 | 0.949 |

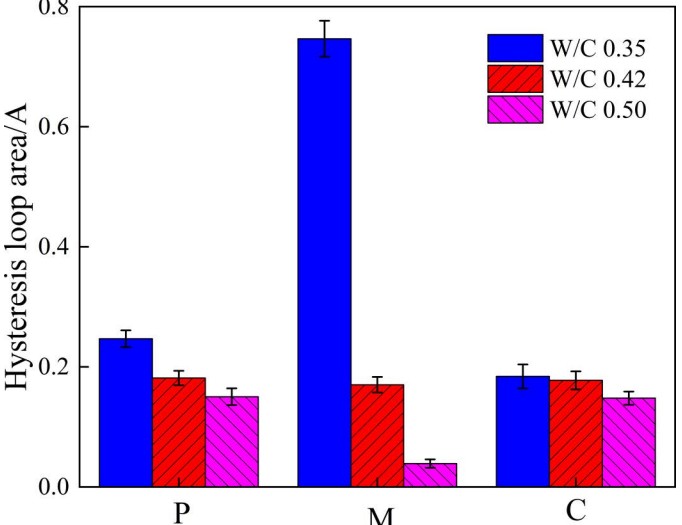

**Fig 10.  Area of hysteresis loop in isothermal adsorption-desorption curve of cement-based materials with different water-cement ratios.**

## 4.1.  Numerical simulation

This study employed a three-dimensional cube model, with the specimen size set at 50 mm × 100 mm × 100 mm. There are six boundaries. To simulate the one-dimensional diffusion situation in actual experiments, two opposite faces were retained as concentration boundaries and flux boundaries, while the other four boundaries were set as no-flux boundaries. Water diffuses one-dimensionally from the erosion surface of the cement-based material, with a temperature of

T = 298.15K. The mix proportion coefficients are $R_1 = 0.35$, $R_2 = 0.42$, and $R_3 = 0.50$. The densities of the paste, mortar, and concrete are $\rho_p = 2050\,kg/m^3$, $\rho_m = 2150\,kg/m^3$, and $\rho_c = 2560\,kg/m^3$, respectively. The density of cement is $\rho = 3100\,kg/m^3$. The porosities of P1, P2, P3, M1, M2, M3, C1, C2, and C3 are 15.4%, 20.05%, 24.93%, 9.0%, 10.25%, 10.73%, 6.86%, 7.67%, and 8.88%, respectively.

Fig 11 shows the mesh division diagram. In this study, triangular meshes were used for mesh division. Triangular meshes have excellent adaptability and numerical stability in simulating concrete erosion, and are particularly suitable for handling complex boundaries, multi-physical field coupling, and heterogeneous material problems.As shown in Fig 12, it is a simulation diagram of water absorption at different times for paste, mortar, and concrete specimens. It can be seen from the figure that as the water absorption age increases, the water content inside the specimens continuously increases. When the water absorption time reaches 15 days, the water content inside the specimens tends to be balanced. After water absorption reaches saturation, the water absorption volume of the paste specimen is greater than that of the mortar specimen, which is greater than that of the concrete specimen, consistent with the experimental results in Section 2.4.

### 4.2. Comparison and analysis of numerical simulation results and experimental results

Fig 13 shows the comparison chart of the numerical simulation results of water absorption and the experimental data. From the figure, it can be seen that the general trend of the experimental results is consistent with that of the simulation results. The simulation results of P1, P2, M1, M3, C1, and C2 are within 10% of the experimental results. The experimental data of P3, M2, and C1 have larger errors when the water absorption time is from 16 hours to 64 hours. The maximum errors are 19%, 30%, and 25% respectively. Analyzing the reasons: The cement-based material is a porous medium with many internal pores. Adding sand and stones will affect the pore structure of the cement-based material, resulting in certain errors between the experimental results and the simulation results. However, the errors are within the controllable range.

### 5. Conclusion

Taking cement-based materials as the research object to study the influence of porosity on the water transmission of cement-based materials, the following conclusions can be obtained:

(1) For cement-based materials with different water-cement ratios, the porosities of P1, P2, P3, M1, M2, M3, C1, C2, and C3 obtained through pressure mercury testing and low-field nuclear magnetic resonance testing were 21.73%, 26.92%, 31.45%, 17.64%, 17.94%, 19.32%, 9.61%, 9.67%, and 12.02% and 15.4%, 20.05%, 24.93%, 9.0%, 10.25%, 10.73%, 6.86%, 7.67%, and 8.88%, respectively. There were certain differences between the porosities measured by pressure mercury testing and low-field nuclear magnetic resonance testing, but the overall trend was

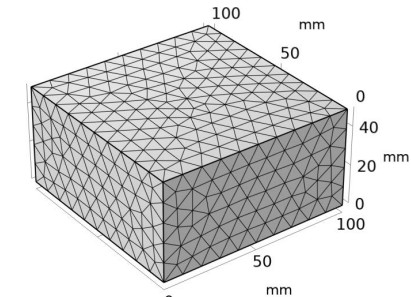

**Fig 11. Meshing.** (a) P,0d, (b) M,0d, (c) C,0d, (d) P,1d, (e) M,1d, (f) C,1d, (g) P,5d, (h) M,5d, (i) C,5d, (j) P,15d, (k) M,15d, (l) C,15d.

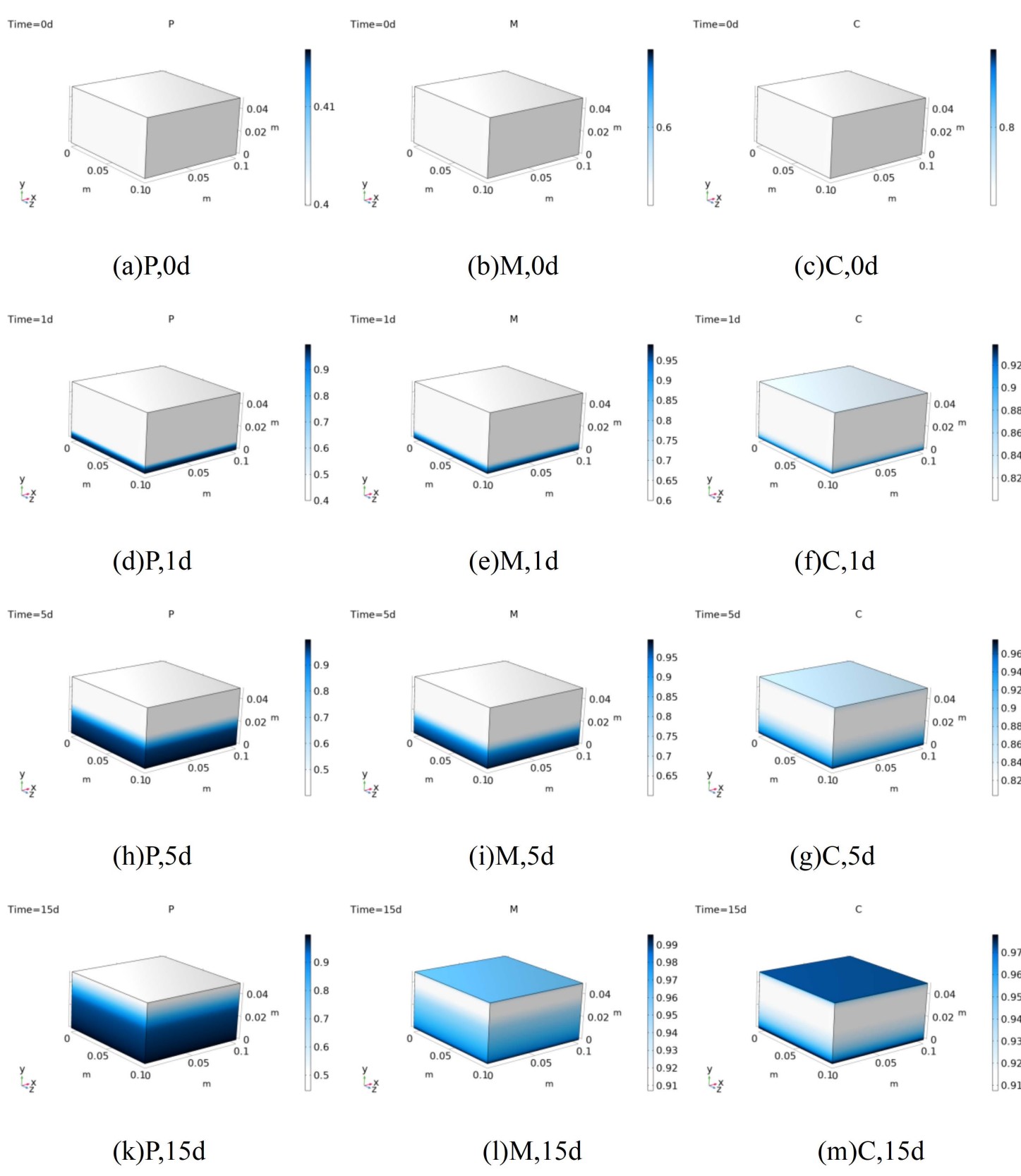

**Fig 12. Water absorption simulation of P, M and C specimens at different time.**

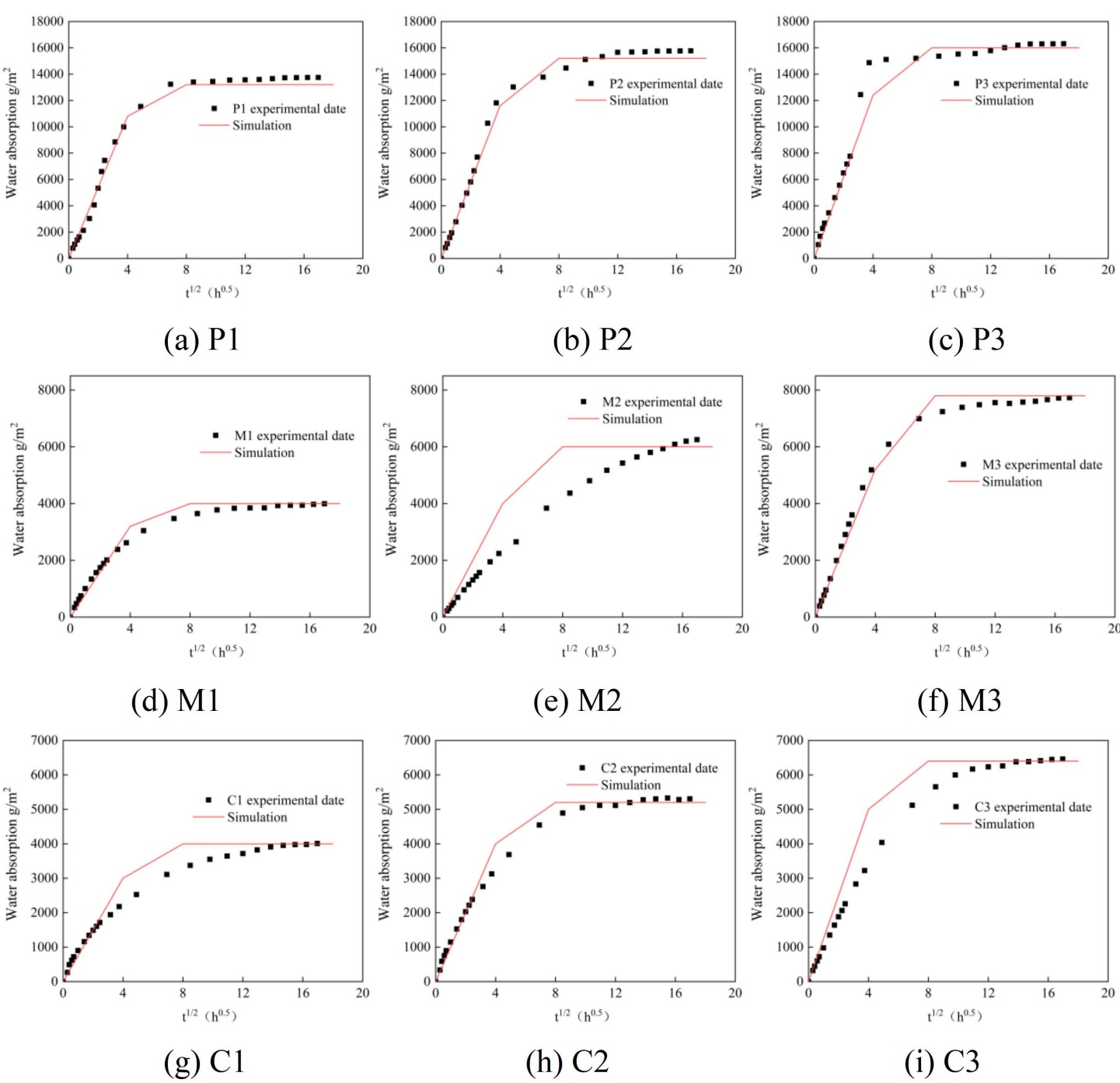

**Fig 13. Comparison of numerical simulation and test data.** (a) P1, (b) P2, (c) P3, (d) M1, (e) M2, (f) M3, (g) C1, (h) C2, (i) C3.

consistent. With the addition of fine aggregate and coarse aggregate, the porosities of mortar and concrete continuously decreased.

(2) For neat paste, mortar, and concrete specimens, as the water-cement ratio increased and sand and gravel were added, the proportion of harmless pores and slightly harmful pores decreased, while the proportion of harmful pores and highly harmful pores increased. The proportions of harmless pores, slightly harmful pores, harmful pores, and highly harmful pores in P3 and P2 were 0.94, 1.91, 2.81, and 0.78 times and 0.97, 2.85, 2.28, and 0.75 times those in P1, respectively; in M3 and M2, they were 0.98, 1.36, 1.33, and 0.93 times and 0.93, 0.68, 0.74, and 1.43 times those in M1, respectively; and in C3 and C2, they were 1.08, 1.51, 0.91, and 0.60 times and 0.97, 1.62, 1.33, and 0.91 times those in C1, respectively.

(3) The water absorption of neat paste, mortar, and concrete with different water-cement ratios varied. The saturated water absorption of P1, P2, P3, M1, M2, M3, C1, C2, and C3 was 12,000, 16,000, 18,000, 4,000, 6,000, 8,000, 3,000, 4,000, and 6,000 $g/m^2$, respectively. Overall, the water absorption of neat paste was greater than that of mortar, which was greater than that of concrete. By adjusting the material ratio, the water absorption performance of cement-based materials can be effectively controlled, thereby improving their durability and service life.

(4) The time required for neat paste, mortar, and concrete to reach saturation was similar, and the porosity had a relatively small impact on the time to reach saturated water absorption. The area of the isothermal adsorption-desorption hysteresis loop of cement-based materials decreased with the increase in water-cement ratio, indicating that the moisture circulation between the humidity environment and the material increased. The numerical simulation results were consistent with the experimental data, further verifying that porosity is a key factor affecting the water transmission of cement-based materials. The higher the porosity, the easier it is for water to enter the material through capillary action.

## Supporting information

**S1 Data.** Figure 1(a) shows the picture taken after the specimen was prepared, Figure 1(b) shows the picture of the specimen's water absorption test, Figure 1(c) shows the picture of the cutting machine used to cut the specimen, and Figure 1(d) shows the picture of the specimen cut by the cutting machine.
(XLSX)

## Author contributions

**Conceptualization:** Dali Zhang.

**Data curation:** Dali Zhang.

**Investigation:** Wenbang Zhu, Yali Cao.

**Methodology:** Wenbang Zhu.

**Project administration:** Ruiming Liu.

**Resources:** Ruiming Liu.

**Software:** Enze Hao, Xiumei Zheng.

**Supervision:** Xinjie Wang, Xiumei Zheng.

**Validation:** Xinjie Wang, Yali Gu.

**Visualization:** Yali Cao, Yali Gu.

**Writing – original draft:** Enze Hao.

**Writing – review & editing:** Yuhang Li.

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
