## [Decision Letter · Decision Letter 0]

Dear Dr. Dali,

Thank you for submitting your manuscript to PLOS ONE. After careful consideration, we feel that it has merit but does not fully meet PLOS ONE’s publication criteria as it currently stands. Therefore, we invite you to submit a revised version of the manuscript that addresses the points raised during the review process.

We look forward to receiving your revised manuscript.

Kind regards,

Ahmed M. Yosri

Academic Editor

PLOS ONE

3. We note that this submission includes NMR spectroscopy data. We would recommend that you include the following information in your methods section or as Supporting Information files:

1) The make/source of the NMR instrument used in your study, as well as the magnetic field strength. For each individual experiment, please also list: the nucleus being measured; the sample concentration; the solvent in which the sample is dissolved and if solvent signal suppression was used; the reference standard and the temperature.

2) A list of the chemical shifts for all compounds characterised by NMR spectroscopy, specifying, where relevant: the chemical shift (δ), the multiplicity and the coupling constants (in Hz), for the appropriate nuclei used for assignment.

3)The full integrated NMR spectrum, clearly labelled with the compound name and chemical structure.

We also strongly encourage authors to provide primary NMR data files, in particular for new compounds which have not been characterised in the existing literature. Authors should provide the acquisition data, FID files and processing parameters for each experiment, clearly labelled with the compound name and identifier, as well as a structure file for each provided dataset. See our list of recommended repositories here: https://journals.plos.org/plosone/s/recommended-repositories "

[The author(s) received no specific funding for this work.]

 [This work was financially supported by the Kashi University-level project (202422930�202422926),2024 Xinjiang Uygur Autonomous Region College Students Innovation and Entrepreneurship Training Program Project (S202410763012).]

Reviewers' comments:

Reviewer's Responses to Questions

**Comments to the Author**

1. Is the manuscript technically sound, and do the data support the conclusions?

Reviewer #1: Yes

Reviewer #2: Partly

Reviewer #3: Yes

Reviewer #4: Yes

Reviewer #5: Partly

2. Has the statistical analysis been performed appropriately and rigorously?

Reviewer #1: Yes

Reviewer #2: N/A

Reviewer #3: No

Reviewer #4: No

Reviewer #5: N/A

3. Have the authors made all data underlying the findings in their manuscript fully available?

Reviewer #1: Yes

Reviewer #2: Yes

Reviewer #3: Yes

Reviewer #4: No

Reviewer #5: Yes

4. Is the manuscript presented in an intelligible fashion and written in standard English?

Reviewer #1: Yes

Reviewer #2: Yes

Reviewer #3: No

Reviewer #4: Yes

Reviewer #5: Yes

Reviewer #1: 1- Although both MIP and NMR techniques were used for porosity characterization, the paper fails to rigorously reconcile the different pore size classification schemes inherent to these methods. The thresholds defining “harmless,” “slightly harmful,” and “harmful” pores are applied equally to both NMR and MIP outputs, despite their differing sensitivities and principles. A critical comparative calibration is needed, particularly when drawing mechanistic conclusions based on porosity proportions.

2- Figure 9 and the accompanying text suggest that an increase in water-cement ratio leads to a “more compact” pore structure based on the decreasing hysteresis loop area. This interpretation contradicts classical understanding, where increased w/c typically produces more porous structures. The decrease in hysteresis area could instead be a result of increased pore connectivity and larger average pore sizes, reducing capillary retention during desorption. This alternative explanation is not considered.

3- The SEM images presented in Figure 6 are used qualitatively to support trends observed in other analyses, but no image processing or quantification is provided. As a result, the SEM section adds limited scientific value and cannot substantiate microstructural claims quantitatively.

4- While the authors attempt to validate their COMSOL simulation model using water absorption data (Section 4), the agreement is shown only qualitatively via contour maps. No statistical or graphical comparison between simulation outputs and experimental absorption curves is offered. This limits the rigor and usefulness of the modeling section.

5- The correlation between porosity (as measured by NMR or MIP) and capillary water absorption is repeatedly emphasized, yet no regression analysis or fitting (e.g., linear, power-law) is provided to quantify these relationships. This omission weakens the claim that porosity directly governs water transport dynamics.

6- While the T2 relaxation times are reported, there is no conversion to pore size distribution using an appropriate model. Furthermore, the authors do not distinguish between bound water and free water peaks, which are critical for understanding transport and retention mechanisms.

7- From MIP, porosity increases with higher w/c ratios, but the proportion of highly harmful pores in paste is lower than in mortar and concrete despite having higher total porosity. This indicates inconsistency or misinterpretation in assigning pore categories. A clearer explanation or cross-validation between MIP and NMR data is needed.

8- Given the scope and content of this paper, it may benefit from considering the following related works:

https://doi.org/10.1007/978-3-031-69626-8_32

https://doi.org/10.1061/(ASCE)MT.1943-5533.0004528

https://doi.org/10.1016/j.matpr.2023.04.103

Reviewer #2: In general, this study is interesting and has the potential to significantly contribute to the field, making it in high demand by the journal's readers. However, there is a significant drawback to the entire manuscript—the presented data is a technical report and not a logically related study. The authors discuss the results obtained by different analysis methods, but the work has no logical coherence and is not scientifically sound. The authors should significantly revise the manuscript and give it a more logical scientific character.

1. It is necessary to clarify what method was used to determine the composition of the cement presented in Table 1

2. Figure 1 and the data presented in it do not carry a deep scientific load and the authors are advised to transfer this figure to the supporting information file. Similarly, Figures 2 and 11.

3. Why are the data in Table 3 presented without statistical processing?

4 The size scale in Figure 6 is unreadable, it needs to be improved.

5. The conclusion also looks like the conclusion of a technical report and not a research manuscript. It needs to be carefully revised

Reviewer #3: The manuscript ‘Research on the Mechanism of Pore Structure on Water Transportation in Cement-based Materials’ investigates the factors influencing the water transport in three different cement-based materials, that is, paste, mortar and concrete, using different experimental techniques and a simulation model. The authors find that porosity decreases by adding to the paste fine and coarse aggregates, making the material more resistant to erosive media and therefore influencing its durability. Furthermore, the water-to-cement ratio is a main factor in compacting the pore structure, an important phenomenon in water retention.

Although the topic is not new and some of the conclusions can be expected, the data shown are interesting and of good quality. The discussion of the results is supported by a numerical simulation model; the content of paragraph 3.5 concerning the absorption-desorption curves is fine. References are complete and updated.

I suggest to publish the paper after the authors have considered the following remarks:

- there is a difference in the linguistic style of the different parts, perhaps due to multiple handwriting. I suggest revising the text by homogenizing the language.

- lines 107-109: since for non-Chinese researchers can be difficult to find these Codes, can you provide a reference?

- line 117: in fact, figure 1d shows the cutting machine, not the samples.

- line 122: can the authors explain the procedure to dry the sample? Did they use the oven at 80°C as before?

- line 130: ‘…according to the standards formulated by the National Technical Committee of Physical Stoichiometry.’ Please, give a reference.

- line 144: ‘m denotes the weight…’: m denotes the mass, not the weight.

- line 168: please, specify the model of SEM used.

- line 170 and following: how was the pressurization achieved? At a constant rate or with other means (e.g. dynamic response)?

- line 177: the SEM and the test sample…: the MIP, not the SEM

- line 185: ‘The pore size measurement range was 0.00002 to 200 μm’: please, control the range: are you able to measure the pore size within 0.02 nm? The Bohr diameter of the hydrogen atom is 0.1 nm.

- line 185: ‘the SEM and the test sample…: the NMR, not the SEM.

- lines 195 and 372: analysis, not analize.

- line 205: in figure 3a, b and c, ‘the volume of mercury intrusion is greater than that of mercury extrusion.’ It seems the contrary: the intrusion lines are always below those of extrusion lines.

- lines 206-208: ‘trend of cumulative mercury intrusion volume reduction is P1 > P2 > P3, M1 > M2 > M3, and C1 > C2 > C3, while the cumulative mercury extrusion volume undergoes negligible change.’ Concerning intrusion, P1 < P2 < P3; furthermore, extrusion volumes are significantly different for P and C samples. Only for M samples they are very similar.

- line 234 and following: the notion of harmless and harmful pores (in relation to durability?) should be defined.

- lines 238 to 250: all the quoted numbers are in my opinion useless, since the proportions are already visible in figure j and k.

- line 290: 90dP: maybe 90 days for C etc.?

- line 296: apparently, a second peak is not visible from figure 4a in the range 6-3000 ms. Can you show it in an inset?

- line 299: ‘indicating that the larger the water-cement ratio, the more the pore size.’ Unclear statement: the larger the size? The higher the number of pores?

- lines 328-348: as noted for MPI (lines 238-250), it is useless to repeat the numbers already shown in figure 5. It makes the text rather confuse.

- line 414: ‘…saturation at 12000’: 12000 g/m^2. Please, specify the units.

- line 415: ‘and that of P3 at 18,000.’ In fact, from the figure it seems that both P2 and P3 saturate at almost 16000 g/m^2.

- line 417-418: ‘The water absorption of the specimens shows that the paste is greater than the mortar.’ Greater in what sense? Please, clarify the sentence.

- Figure 9: can the authors add the error bars? Particularly for C samples it should be useful, in order to support the claim that the area of hysteresis loop decreases by increasing the water-to-cement ratio.

- line 546: ‘When the water absorption time reaches 10 days…’ This is not shown in the figure.

Reviewer #4: This work presents experimental studies on water absorption/adsorption in paste, mortar, and concrete with three water-binder ratios, supported by CFD simulations. While the results are of potential interest to the civil engineering community, the manuscript is overly lengthy and reads more like a preliminary report than a ready-to-submit journal article. Therefore, it is not of sufficient quality to motivate publication for now.

My major/minor concerns are the following:

1. Sections 3.1–3.3 should be consolidated into a single subsection with side-by-side comparisons and a schematic diagram of pore evolution to enhance readability and synthesis.

2. The lengthy descriptive passages (e.g., Lines 232–268, 290–320, 325–368, 412–440) should be streamlined. Some observations could be replaced with schematics or summarized more concisely.

3. Avoid redundant subsections like "Analysis of the reasons" (Lines 258, 313)—integrate explanations directly into results.

4. The CFD simulation setup is not described in the Methods section, leaving readers unprepared for Section 4. Provide details on boundary conditions, meshing, and validation.

5. Equation (2) (Line 407) lacks derivation or citation. Justify its reasonable use and include a derivation in the Supporting Material.

6. Clarify the definition of A (water absorption surface area)—is this cross-sectional area or pore surface area? (Line 408)

7. Figure 3

7.1) It is hard to read the graphs due to inconsistent color and symbols. Let me pick P1 as an example for explanation. The intrusion-extrusion curves of P1 (Figure 3(a)) should be with the same color (black) and differentiate intrusion-extrusion with symbols/line styles. Also, use the same color for P1/M1/C1.

7.2) Figures 3(g-i): Clarify the y-axis unit for cumulative mercury volume.

7.3) Figures 3(j-k): What is the number in x-axis? Is it a sample code?

7.4) Check the consistency of figure information.

8. Lines 206-208: The authors should re-examine on the ranking of cumulative mercury intrusion volume and the explanation of cumulative mercury extrusion volume.

9. Line 105: The abbreviation of paste, mortar, and concrete should be specified here.

10. Table 1: Is chemical composition of cement in mass fraction or mass percentage?

11. Table 2: Capitalize “Superplasticizer”.

12. Line 130: Cite the source for the National Technical Committee of Physical Stoichiometry.

13. Lines 168-169, 177-178, 185-186: Please check the description.

14. Line 195: MIP analyze or MIP analysis?

Line 372: SEM analyze or SEM analysis?

15. Lines 216-220: The interpretation of mercury intrusion-extrusion behavior requires clarification. The claim that “The insignificant change in the cumulative mercury volume” contradicts the visible hysteresis in Figure 3. While the extrusion curve does not mirror the intrusion trend, the disparity itself is significant and should be discussed as evidence of pore entrapment or ink-bottle effects. Moreover, the assertion that pore connectivity is "stable" (i.e., non-deformable) lacks justification. MIP hysteresis alone cannot confirm this—connectivity could evolve due to mechanical deformation or expansion of pore.

16. Figure 5: number?

17. Lines 414-417: Unit of water absorption capacity?

18. Figure 7: Can we plot the graphs with x-axis of time (h)? I do not see the specific reason in the discussion why to select the square root of time in x-axis.

19. Figure 8: The authors have plotted the equilibrium absorption-desorption isotherms of water in cement-based materials. However, in determining the area of the hysteresis loop, increasing the number of data points for the hysteresis loop may improve accuracy. Moreover, the lower closure point of the hysteresis loop should not artificially be zero in typical. Therefore, I suggest that the authors add a greater number of points. Also, the area of the hysteresis loop can be better computed through numerical integration to reduce numerical truncation errors rather than the curve fitting with simple cubic equation at absorption and desorption branches.

20. The first conclusion (Section 5) is overly broad and states a well-known fact. Focus on specific findings from this study

Reviewer #5: Comments from Reviewer

Manuscript ID: PONE-D-25-11920_reviewer.pdf

Title: Research on the Mechanism of Pore Structure on Water Transportation in Cement-based Materials

The authors investigated the water transport mechanism in cement-based materials with different water-cement ratios. The influence of the pore structure, three types of specimens with different water-cement ratios (paste, mortar, and concrete) was investigated. The authors investigated capillary water absorption and isothermal adsorption-desorption water transport. Additionally, the pore changes and porosity were analyzed. The obtained results show that the water transport speed of the paste is greater than that of the mortar and concrete, which is in good agreement with the experimental results.

The current form's presentation of methods and scientific results is unsatisfactory for publication in the PLOS ONE journal. The article is interesting but poorly prepared. Some comments apply to the entire article. Please take this into account when making corrections. The minor and significant drawbacks to be addressed can be specified as follows:

Minor comments:

1. (See 2.2) 2.1. Raw Materials - 2.1. Raw materials

2. 2.3 Test Methodology - 2.3 Test methodology

3. Tab. 2. Superplasticizer - Superplasticizer

4. Line 144. Where - where

5. Tab. 3. humidness - Humidness

6. 5.conclusion - 5. Conclusion

Major comments:

1. Fig. 3, panels (a) – (c) Why were lower pressure measurements not performed for some samples?

2. The authors were interested in the range of pores up to 200 nm. Why did the authors not perform low-temperature measurements of nitrogen adsorption? Mercury porosimetry is a method that is burdened with error. A combination of these two methods would be very desirable.

3. All figures - axis descriptions. Sometimes titles start with a lowercase letter, and sometimes with an uppercase letter. Please standardize.

4. Fig. 6. Very poorly visible information about the resolutions.

5. Figs. 7 and 8. After the adsorption/desorption cycle, was another cycle of measurements performed on the same sample?

6. Tab. 4. Too high precision of the given values.

Sincerely,

The reviewer.

**Do you want your identity to be public for this peer review?** For information about this choice, including consent withdrawal, please see our Privacy Policy

Reviewer #1: No

Reviewer #2: No

Reviewer #3: No

Reviewer #4: No

Reviewer #5: **Yes: ** Piotr A. Gauden

---

## [Author Response · Author response to Decision Letter 1]

9 Jun 2025

Responses to Reviewers’ Comments for the paper entitled

Study on chloride attack resistance of concrete with lithium slag content

Buildings

Manuscript ID: buildings-PONE-D-25-11920

by Enze HAO Yuhang LI,Dali ZHANG Wenbang ZHU Ruiming LIU Xinjie WANG Yali CAO,Yali GU, Xiumei ZHENG

The authors wish to thank the editor and five reviewers for their very thorough, insightful, and constructive reviews. All comments have been addressed in the revised version of the manuscript. Original comments of each reviewer have been placed in boxes in this document and specific responses have been placed in italics for ease of reference. In the responses, all the numbers of figures, tables, and lines refer to the revised manuscript, unless otherwise specified.

Reviewer #1:

The authors thank the reviewer for the valuable and careful comments. All of the following comments have been addressed in the revised manuscript.

In order to improve the quality of this paper, the full test has been checked and corrected in the revised manuscript.

1 Although both MIP and NMR techniques were used for porosity characterization, the paper fails to rigorously reconcile the different pore size classification schemes inherent to these methods. The thresholds defining “harmless,” “slightly harmful,” and “harmful” pores are applied equally to both NMR and MIP outputs, despite their differing sensitivities and principles. A critical comparative calibration is needed, particularly when drawing mechanistic conclusions based on porosity proportions.

The authors are grateful to the reviewer's precious comment.In this study, the pores are classified into four categories following the classification proposed by Academician Wu Zhongwei. The first category is harmless pores, with diameters ranging from 0 nm to 20 nm. These pores are usually closed or isolated and do not form a connected network. They hardly reduce the mechanical strength of the material and have no significant contribution to permeability. The second category is slightly harmful pores, with diameters ranging from 20 nm to 50 nm. These pores may form local connected paths, but the overall permeability is relatively low. They slightly weaken the strength and allow a small amount of water or ions to slowly penetrate, but the threat to durability is limited. The third category is harmful pores, with diameters ranging from 50 nm to 200 nm. These pores are prone to form continuous channels, significantly increasing the material's permeability, obviously reducing the density and mechanical strength, and accelerating the invasion of harmful substances such as chloride ions and sulfates. The fourth category is highly harmful pores, with diameters greater than 200 nm. These are large pores or crack-level pores, usually caused by construction defects or shrinkage stress. They severely damage the material's strength and impermeability, becoming stress concentration points and crack propagation paths, and significantly shorten the service life of the structure. The purpose of using MIP and NMR to test the pores and adopting the same pore size distribution is to control the variables and better compare the MIP test with the NMR test. Lines 241-259 and 350-365 of the article.

2 Figure 9 and the accompanying text suggest that an increase in water-cement ratio leads to a “more compact” pore structure based on the decreasing hysteresis loop area. This interpretation contradicts classical understanding, where increased w/c typically produces more porous structures. The decrease in hysteresis area could instead be a result of increased pore connectivity and larger average pore sizes, reducing capillary retention during desorption. This alternative explanation is not considered.

Thanks to the reviewer for raising a very important problem.It has been considered that due to the increase in pore connectivity and the enlargement of the average pore diameter, the capillary retention during the desorption process can be reduced. This is discussed in lines 524 to 527 of the article.

3 The SEM images presented in Figure 6 are used qualitatively to support trends observed in other analyses, but no image processing or quantification is provided. As a result, the SEM section adds limited scientific value and cannot substantiate microstructural claims quantitatively.

The authors are grateful to the reviewer's precious comment.The SEM image in Figure 6 was processed, and this was done in lines 387 to 394 of the article.

4 While the authors attempt to validate their COMSOL simulation model using water absorption data (Section 4), the agreement is shown only qualitatively via contour maps. No statistical or graphical comparison between simulation outputs and experimental absorption curves is offered. This limits the rigor and usefulness of the modeling section.

Thanks. In Section 4.2 of the article, a comparison analysis between experiments and simulations has been added, starting from line 572 to line 591.

5 The correlation between porosity (as measured by NMR or MIP) and capillary water absorption is repeatedly emphasized, yet no regression analysis or fitting (e.g., linear, power-law) is provided to quantify these relationships. This omission weakens the claim that porosity directly governs water transport dynamics.

Thanks to the reviewer for raising a very important problem. To enhance the relationship between porosity and water absorption, a fitting graph of porosity and water absorption for MIP and NMR tests was added in the 452 to 466 lines of the article.

6 While the T2 relaxation times are reported, there is no conversion to pore size distribution using an appropriate model. Furthermore, the authors do not distinguish between bound water and free water peaks, which are critical for understanding transport and retention mechanisms.

Thanks for the reviewer's question. In the numerical simulation of this study, only porosity was considered, while the differences in pore distribution, free water and bound water transmission were not taken into account. There was indeed a certain error when comparing the simulation results with the experiments, but the error was within a controllable range and the overall trend was consistent. Thank you for the expert's opinion. We will make improvements in the later research.In lines 572 to 591 of the article.

7 From MIP, porosity increases with higher w/c ratios, but the proportion of highly harmful pores in paste is lower than in mortar and concrete despite having higher total porosity. This indicates inconsistency or misinterpretation in assigning pore categories. A clearer explanation or cross-validation between MIP and NMR data is needed.

Thanks to the reviewer for raising a very important problem.A comparative analysis of nuclear magnetic resonance and pressure generation has been added, starting from line 350 to line 365 of the article.

8 Given the scope and content of this paper, it may benefit from considering the following related works:

https://doi.org/10.1007/978-3-031-69626-8_32

https://doi.org/10.1061/(ASCE)MT.1943-5533.0004528

https://doi.org/10.1016/j.matpr.2023.04.103

The authors are grateful to the reviewer's precious comment.This study has referred to the above three articles and added citations, which can be found in lines 672-686 of the article.

Reviewer #2:

The authors thank the reviewer for the valuable and careful comments. All of the following comments have been addressed in the revised manuscript.

In order to improve the quality of this paper, the full test has been checked and corrected in the revised manuscript.

1.It is necessary to clarify what method was used to determine the composition of the cement presented in Table 1

Thanks for pointing out. It has been stated in lines 114-115 of the article what method was used to determine the composition of the cement.

2. Figure 1 and the data presented in it do not carry a deep scientific load and the authors are advised to transfer this figure to the supporting information file. Similarly, Figures 2 and 11.

Thanks for the kind remind from the reviewer. Figure 1 shows the sample fabrication and testing process, while Figure 2 presents the Microscopic instruments and test samples. The contents of Figure 1 and Figure 2 have been simplified, some pictures have been deleted, and some pictures have been placed in the attachment. Figure 12: Water absorption simulation of P, M and C specimens at different times. In Section 4.2 of the article, an experimental and simulation comparison analysis has been added, and Figure 12 has enhanced the correlation between the experiment and the simulation. This is in lines 169-174, 202-208, and563-571 of the article.

3.Why are the data in Table 3 presented without statistical processing?

Many thanks for the precious comment. Table 3 Humidity corresponding to saturated salt solution, Refer to [42] Hygrothermal performance of building materials and products - Determination of moisture content by drying at elevated temperature; Amendment 1: ISO 12570 AMD 1-2013 [S], 2013. (Translated from Chinese) Data obtained from the literature.This is in lines 175 and 773-775 of the article.

4.The size scale in Figure 6 is unreadable, it needs to be improved.

Thanks for the reviewer's question. The SEM images of cement-based materials with different water-cement ratios in Figure 6 have been replaced. Higher-precision images have been used, and the size and proportion are clearly visible. These images can be found in lines 387-395 of the article.

5.The conclusion also looks like the conclusion of a technical report and not a research manuscript. It needs to be carefully revised

Thanks for the reviewer's question. The conclusion of the article has been revised and can be found on lines 596 to 629.

Reviewer #3:

The authors thank the reviewer for the valuable and careful comments. All of the following comments have been addressed in the revised manuscript.

In order to improve the quality of this paper, the full test has been checked and corrected in the revised manuscript.

1.lines 107-109: since for non-Chinese researchers can be difficult to find these Codes, can you provide a reference?

Thanks for the reviewer's question.Reference materials have been provided, and they can be found on pages 769 to 772 of the article.

2.line 117: in fact, figure 1d shows the cutting machine, not the samples.

Thanks for the reviewer's question. The text has been revised. It has been changed at line 137 of the article.

3.line 122: can the authors explain the procedure to dry the sample? Did they use the oven at 80°C as before?

Many thanks for the precious comment.The water absorption experiment reached the specified time points, with durations of 5 minutes, 10 minutes, 20 minutes, 30 minutes, 1 hour, 2 hours, 3 hours, 4 hours, 5 hours, 6 hours, 10 hours, 14 hours, and 24 hours respectively. After drying the surface moisture of the specimens, the weights were measured. The weights were measured without using a drying oven. In lines 141-145 of the article.

4.line 130: ‘…according to the standards formulated by the National Technical Committee of Physical Stoichiometry.’ Please, give a reference.

Thanks for the kind remind from the reviewer.References are provided on lines 150-151 of the article.

5.line 144: ‘m denotes the weight…’: m denotes the mass, not the weight.

Thanks for pointing out.The modification is made. "m" represents mass, and it is located in line 176 of the article.

6.line 168: please, specify the model of SEM used.

Thanks.The manufacturers and models of SEM that have been added are listed in lines 177-179 of the article.

7.line 170 and following: how was the pressurization achieved? At a constant rate or with other means (e.g. dynamic response)?

Thanks.This study employed the AutoPore IV 9500 pore measurement instrument produced by the American company Micromeritics. The pore measurements within a specific pore size range were achieved through staged pressure application (low-pressure pneumatic + high-pressure hydraulic). It is necessary to strictly control the pressure gradient, equilibrium time, and safety precautions. This is stated in lines 188-189 of the article.

8. line 177: the SEM and the test sample…: the MIP, not the SEM

Thanks for the reviewer's question.This sentence has been deleted.

9.line 185: ‘The pore size measurement range was 0.00002 to 200 μm’: please, control the range: are you able to measure the pore size within 0.02 nm? The Bohr diameter of the hydrogen atom is 0.1 nm.

Thanks.The aperture measurement range has been revised to 0.0002 - 200 μm, as stated on line 201 of the article.

10. line 185: ‘the SEM and the test sample…: the NMR, not the SEM.

Thanks for the reviewer's question.This sentence has been deleted.

11. lines 195 and 372: analysis, not analize.

Thanks.The text has been revised, at lines 211 and 366 of the article.

12. line 205: in figure 3a, b and c, ‘the volume of mercury intrusion is greater than that of mercury extrusion.’ It seems the contrary: the intrusion lines are always below those of extrusion lines.

Thanks for pointing out.The modifications have been made in lines 221-222 of the article.

13. lines 206-208: ‘trend of cumulative mercury intrusion volume reduction is P1 > P2 > P3, M1 > M2 > M3, and C1 > C2 > C3, while the cumulative mercury extrusion volume undergoes negligible change.’ Concerning intrusion, P1 < P2 < P3; furthermore, extrusion volumes are significantly different for P and C samples. Only for M samples they are very similar.

Many thanks for the precious comment.During the drawing process of the Mercury curve of P, an error occurred and the line types of P1 and P3 were reversed. When drawing the mercury curve of C, due to the addition of coarse aggregates in the concrete, it had a certain impact on the connectivity of the concrete interior, resulting in some errors in the mercury insertion and withdrawal tests of C2 and C3 specimens. However, the overall trend was generally in line with the pattern. At line 221-227 and 276 of the article.

14. line 234 and following: the notion of harmless and harmful pores (in relation to durability?) should be defined.

Thanks for the reviewer's question.The definitions of harmless holes, slightly harmful holes, harmful holes, and highly harmful holes have been added. They are located in lines 241 to 259 of the article.

15. lines 238 to 250: all the quoted numbers are in my opinion useless, since the proportions are already visible in figure j and k.

Thanks for the reviewer's question.The description in the original text has been deleted.

16. line 290: 90dP: maybe 90 days for C etc.?

Thanks.For the specimens of mortar, concrete and concrete mixtures, they were all cured for 90 days. This was done to allow the cement to fully hydrate, prevent errors caused by the cement hydration process, and to control the variables.

17. line 296: apparently, a second peak is not visible from figure 4a in the range 6-3000 ms. Can you show it in an inset?

Thanks for the reviewer's question.In Figure 4(a), there is only one main peak, and an illustration has been added, on line 320 of the article.

18. line 299: ‘indicating that the larger the water-cement ratio, the more the pore size.’ Unclear statement: the larger the size? The higher the number of pores?

Thanks.The changes have been made in lines 302-304 of the article.

19. lines 328-348: as noted for MPI (lines 238-250), it is useless to repeat the numbers already shown in figure 5. It makes the text rather confuse.

Thanks.The description in the original text has been deleted and modified. These changes have been made in lines 324-346 of the article.

20. line 414: ‘…saturation at 12000’: 12000 g/m2. Please, specify the units.

Thanks.The unit "g/m2" has been added in pages 416-419 of the article.

21. line 415: ‘and that of P3 at 18,000.’ In fact, from the figure it seems that both P2 and P3 saturate at almost 16000 g/m2.

Thanks for the kind remind from the reviewer.The modifications have been made in lines 416-417 of the article. P2 and P3 reached saturation at approximately 15700 g/

---

## [Decision Letter · Decision Letter 1]

Research on the Mechanism of Pore Structure on Water Transportation in Cement-based Materials

PONE-D-25-11920R1

Dear Dr. Dali,

We’re pleased to inform you that your manuscript has been judged scientifically suitable for publication and will be formally accepted for publication once it meets all outstanding technical requirements.

Kind regards,

Anwar Khitab

Academic Editor

PLOS ONE

Additional Editor Comments (optional):

Reviewers' comments:

Reviewer's Responses to Questions

**Comments to the Author**

Reviewer #3: All comments have been addressed

Reviewer #4: All comments have been addressed

2. Is the manuscript technically sound, and do the data support the conclusions?

Reviewer #3: Yes

Reviewer #4: Yes

3. Has the statistical analysis been performed appropriately and rigorously?

Reviewer #3: Yes

Reviewer #4: Yes

4. Have the authors made all data underlying the findings in their manuscript fully available?

Reviewer #3: Yes

Reviewer #4: Yes

5. Is the manuscript presented in an intelligible fashion and written in standard English?

Reviewer #3: Yes

Reviewer #4: Yes

Reviewer #3: (No Response)

Reviewer #4: The authors have partially responded my comments with reasonable facts from what they found. I thereby accept the current version of the revised manuscript.

**Do you want your identity to be public for this peer review?** For information about this choice, including consent withdrawal, please see our Privacy Policy

Reviewer #3: No

Reviewer #4: No

---

## [Editor Report · Acceptance letter]

PONE-D-25-11920R1

PLOS ONE

Dear Dr. Zhang,

I'm pleased to inform you that your manuscript has been deemed suitable for publication in PLOS ONE. Congratulations! Your manuscript is now being handed over to our production team.

Kind regards,

on behalf of

Professor Anwar Khitab

Academic Editor

PLOS ONE